# Framework for Climate Change Adaptation of Agriculture and Forestry in Mediterranean Climate Regions

André Vizinho [1,*], David Avelar [1], Cristina Branquinho [1], Tiago Capela Lourenço [1], Silvia Carvalho [1], Alice Nunes [1], Leonor Sucena-Paiva [1], Hugo Oliveira [1], Ana Lúcia Fonseca [1], Filipe Duarte Santos [1], Maria José Roxo [2] and Gil Penha-Lopes [1]

[1] Centre for Ecology, Evolution and Environmental Changes (ce3c), Faculdade de Ciências da Universidade de Lisboa, Campo Grande, Bloco C2, Piso 5, 1749-016 Lisboa, Portugal; dnavelar@fc.ul.pt (D.A.); cmbranquinho@fc.ul.pt (C.B.); tcapela@fc.ul.pt (T.C.L.); sccarvalho@fc.ul.pt (S.C.); amanunes@fc.ul.pt (A.N.); mlpaiva@fc.ul.pt (L.S.-P.); hfoliveira@fc.ul.pt (H.O.); alfonseca@fc.ul.pt (A.L.F.); fdsantos@fc.ul.pt (F.D.S.); gppenha-lopes@fc.ul.pt (G.P.-L.)

[2] Faculdade de Ciências Sociais e Humanas, Universidade Nova de Lisboa, Avenida de Berna, 26-C, 1069-061 Lisboa, Portugal; mj.roxo@fcsh.unl.pt

* Correspondence: andrevizinho@fc.ul.pt; Tel.: +351-965615379

**Abstract:** Planning the adaptation of agriculture and forestry landscapes to climate change remains challenging due to the need for integrating substantial amounts of information. This information ranges from climate scenarios, geographical site information, socio-economic data and several possible adaptation measures. Thus, there is an urgent need to have a framework that is capable of organizing adaptation strategies and measures in the agriculture and forestry sectors in Mediterranean climatic regions. Additionally, this framework should provide a cause effect relation with climate vulnerability to adequately support the development of adaptation planning at municipal and local (farm) level. In this context, we propose to test and evaluate a framework for climate adaptation of the agriculture and forestry sectors, based on the local causal-effect relation between adaptation strategies and measures and the level of vulnerability reduction achieved for Mediterranean areas. The framework was developed based on the combination of the DPSIR (Driving forces, Pressures, State, Impacts, Responses) and Vulnerability frameworks and reviewed 162 practical adaptation measures, further organized into strategies, complemented by a set of efficacy indicators. The framework was tested with 70 stakeholders in six stakeholder workshops for the planning of two farms and one municipal climate adaptation study, that are now in actual implementation and monitoring. The framework is composed by a set of eight adaptation strategies in which adaptation measures are clustered and assessed using efficacy indicators. In the evaluation of the adaptation framework, 96% of stakeholders considered its content as good or very good and 89% considered the final outcomes as good or very good. Finally, the framework was also used to assess and compare the adaptation strategies and measures presented in the climate adaptation plans of the three case studies. On average, 52.2% of the adaptation measures selected by the three case studies are dedicated to Ecosystem Resilience, 30.9% to Adaptive Capacity, 9.1% to Microclimates, 7.4% to Protection, and 0.3% to Mitigation strategies. This framework was considered effective in supporting adaptation planning at farm and municipal levels and useful to assess and compare adaptation plans in the frame of vulnerability reduction. Future studies can further contribute to support adaptation planning in these sectors by using, developing and streamlining this framework to additional and different socio-ecological contexts.

**Keywords:** climate change adaptation; landscape planning; farm adaptation; municipal adaptation; agroforestry; efficacy; decision making

## 1. Introduction

The Paris Agreement sets out a target to limit the increase in global mean temperature to well below 2 °C above pre-industrial levels and to pursue efforts to limit that increase to 1.5 °C [1]. However, the latest projections, according to the current policies, point to substantially higher levels of warming unless radical actions to cut greenhouse gas (GHG) emissions are set in motion [2]. Scenarios of climate change for the Mediterranean region until the end of the XXI century [3] project an increasing suite of climate risks, including loss of ecosystem services, desertification and land degradation, migration of animals and the degradation of extensive areas of forests and agroforestry systems [4]. These cause severe economic, social and environmental costs [5–7] which make climate adaptation essential, particularly in Mediterranean Climate Regions [8,9].

Planning for climate change adaptation must bridge across science and society, integrate complexity and deal with the challenge of managing and communicating complexity and uncertainty [10–14]. The uncertainty about the future climate comes not only from the GHG emission models but also from climate model limitations. Uncertainty increases with the downscaling of global and regional climate models to local models, largely due to variations in topography, soil, winds, water, ecosystem type, etc. [13,15,16].

In the sector of agriculture and forestry adaptation measures may focus on: (i) the crop (the adaptation may consist of changing or improving the crop/species/variety in itself); (ii) the cropping system (includes the management practices and techniques as well as crop rotation and timings); and (iii) the farming system (includes the farmer and, therefore, the adaptation capacity, the market, regulation, economic incentives, protection mechanisms and information) [8]. Thus, adaption in this sector requires knowledge on the climate sensitivity of each crop, species and variety under different soil conditions (e.g., water retention, nutrients, organic matter and structure, among others), exposure to climate variables throughout the year (e.g., solar radiation, humidity, rainfall, temperature), interactions with other species (e.g., crops, biodiversity, pests), land uses and the interdependency with the agricultural techniques used [17,18].

Although, adaptation measures can be effective for several crops, beyond certain thresholds of climate change, adaptation measures must be complemented with more systemic changes such as diversification of production systems and livelihoods, that increases system resilience [19]. These more systemic adaptation actions address other climate risks such as changes in markets, which will increase due to the impacts in agriculture productions in many regions [19]. To facilitate the integration of this complexity in adaptation planning, Howden et al. conclude that "a crucial component of this approach is the implementation of adaptation assessment frameworks that are relevant, robust, and easily operated by all stakeholders, practitioners, policymakers, and scientists" [19]. In addition, due to the risk of maladaptation, several studies point out a clear need for adaptation frameworks that can bring, to the top of the planning agenda, the full overview of the adaptation strategies, namely in combination with mitigation [20–22]. The need for such a framework was also identified in our participatory research, as we propose stakeholders to analyse and discuss the adaptation measures for the agriculture and forestry of the region in study. The vast list of adaptation measures and, most importantly, the degree to which some fit in others or are effective by addressing different aspects of vulnerability, showed the stakeholders and the authors of this study the need for a sector specific adaptation framework. After reviewing the adaptation frameworks for agriculture and forestry we found no framework that is capable of organising the hierarchy of the adaptation strategies and measures in a robust structure that is based on cause effect relations and, is, at the same time, easily used to support real-world decision making (see Section 1.1).

The problem that this study aims to address is thus the lack of an adaptation framework that can adequately support the adaptation planning in the sector of agriculture and forestry, namely at the farm and local spatial planning level. To address this problem, we need a conceptual, hierarchical, causal, and functional organization of adaptation strategies and measures. Furthermore, in order to assess, evaluate and choose adaptation measures

for specific agriculture and forestry contexts a set of indicators of efficacy is required. The aim of this study is to propose, test and evaluate a framework for climate adaptation of the agriculture and forestry sectors, that is able to support farm-level and municipal-level adaptation planning in Mediterranean areas.

### 1.1. Adaptation Frameworks

The Driving forces, Pressure, State, Impact, Responses (*DPSIR*) framework [23,24], known and used for a long time in the analysis of environmental problems, defines a chain of causal links that help to decide if we want to act on the causes or on the consequences of an impact. It starts with *Driving forces* (e.g., human activities and, in this case, greenhouse gases emissions), followed by *Pressures* (e.g., climate variables, emissions, waste, pollution), *State* (e.g., physical, chemical, biological, ecological elements and functioning of the system) and *Impacts* (on ecosystems, health, functions), eventually leading to *Responses* (in this case, adaptation measures). While this framework continues to provide a cause effect rationale for analysing environmental problems and responses, several Climate Change Adaptation (CCA) frameworks have, meanwhile, been developed to support adaptation planning. Bours et al. have reported and analysed 16 of these frameworks that apply to different sectors and scales, some more theoretical and others more practice oriented. Despite the large number of frameworks, in their analysis they still conclude that "the evidence base informing CCA is still fragmentary and nascent" and that "they have shown only moderate practical effect in reducing vulnerabilities" [25]. Regarding indicators, these authors state that an effort must be made to harvest innovative indicators more directed to applied research [25].

The *Vulnerability Framework* [26] defines adaptation as the actions that reduce the vulnerability to climate change potential impacts. The potential impacts are a function of the exposure to climate variables and the sensitivity of the system. The vulnerability is a function of the potential impacts with the capacity to adapt. The *UKCIP Adaptation Wizard* is based on identifying present and future vulnerabilities, identifying criteria for decision making and then identifying and evaluating adaptation measures to reduce those vulnerabilities [25]. The *Adaptation Pathways* [27,28] uses the concept of tipping points when a certain adaptation action is no longer effective. When a tipping point is reached, another adaptation action must be implemented. In this framework, the adaptation pathways and tipping points map is only developed after the identification of the vulnerabilities and the identification and quantification of the efficacy of adaptation actions. Thus, to use this framework in agriculture and forestry sectors a comparison of the efficacy of adaptation measures must be available *a priori*.

There are several other frameworks that, due to their nature, partially overlap with the objectives of the *Climate Adaptation frameworks*. The *Disaster Risk Reduction (DRR)* is a framework that focuses on extreme events like fires, droughts, storms, and floods, including therefore the impacts of climate change. Climate adaptation frameworks focus on changes in average values of precipitation and temperature and their distribution over time, which include extreme events [29].

### 1.2. Adaptation Frameworks for Agriculture and Forestry Sectors

In the context of agriculture and forestry, Mitter et al. [30] classified the process of adaptation into *3 types of implementation*: (a) incremental, (b) systemic or (c) transformational separated. Robert et al. [31] draw attention to *the temporal scale of adaptation* and they found that 70% of the adaptation studies focus only in one of these time scales of adaptation (day, or season or long-term), highlighting the need to integrate these dimensions in a multi-temporal scale approach.

Some authors developed more specific *frameworks for the agriculture and forest sector*. Smit and Skinner [18] characterized the adaptation measures according *to aim, timing and duration, scale, responsibility and form* and organized adaptation measures into the following types: (i) technological developments; (ii) government programs and insurance;

(iii) farm production practices; and (iv) farm financial management. On the other hand, Kurukulasuriya and Rosenthal [32] developed *typologies more related to the time when the effects of the adaptation measures will be observed:* (i) short term adaptations; (ii) long term adaptations; and (iii) adaptations irrespective of the temporal dimension of climate impacts. Hernández-Morcillo et al. [22], prioritized the agroforestry measures for adaptation and mitigation *according to their perceived performance.* Focusing on mitigation, they proposed several measures for sequestering carbon or reducing greenhouse gases, whereas focusing on adaptation they proposed measures for enhancing resilience or reducing threats.

The *Mediterranean forest sector* alone was already the object of different adaptation frameworks. Regato et al. [33] divided the adaptation measures by the typology of actions: (i) changes in tree species composition; (ii) conservation/restoration of biotic dispersal vectors; (iii) changes in silvicultural practices; (iv) changes in soil management practices; (v) changes in forestry guidelines; and (vi) landscape adaptation options. Vilà-Cabrera et al. [34] defined three general objectives for the adaptation of Mediterranean forests: (1) decrease disturbance risk; (2) increase resistance to disturbance and (3) promote recovery after disturbance. Furthermore, they defined five adaptation strategies: (a) reduction in stand density; (b) management of the understory; (c) promoting mixed forests; (d) changing species or genetic composition; and (e) promoting spatial heterogeneity at the landscape level.

The adaptation frameworks, classification objectives and components, previously presented and summarized in Table 1, show that none of the components or categories used to organize the adaptation measures are similar or integrate each other and, therefore, add complexity to the adaptation planner. Although relevant to understand and categorize adaptation actions in agriculture and forestry, they do not organize the adaptation measures in relation to vulnerability reduction. Instead, they organize the adaptation components in relation to: types of implementation (Mitter et al. [30]); temporal scale (Robert et al. [31]); aim, timing and duration, scale, responsibility and form (Smit and Skinner [18]); the time needed for effects to be observed (Kurukulasuriya and Rosenthal [32]); their perceived performance (Hernández-Morcillo et al. [22]); and the typology of actions (Regato et al. [33]) or general objectives (Vilà-Cabrera et al. [34]).

**Table 1.** Climate adaptation frameworks for agriculture and forestry. Each framework categorizes adaptation measures according to different classification objectives. The different components presented by the different frameworks, are shown in columns Component 1 to Component 6 and illustrate the complexity present in planning adaptation in this sector.

| Adaptation Framework | Classification Objective | Component 1 | Component 2 | Component 3 | Component 4 | Component 5 | Component 6 |
|---|---|---|---|---|---|---|---|
| Hernandéz-Morcillo et al. (2018) [30] | Perceived performance | Mitigation (sequestering carbon and reducing green-house gases) | Adaptation (enhancing resilience and reducing threats) | - | - | - | - |
| Vilà-Cabrera et al. (2018) [34] | Adaptation Strategies | Reduction in stand density | Management of the understory | Promoting mixed forests | Changing species or genetic composition | Promoting spatial heterogeneity at the landscape level | - |
| Vilà-Cabrera et al. (2018) [34] | Adaptation Objectives | Decrease risk | Increase resistance | Promote recovery | - | - | - |
| Mitter et al. (2018) [30] | Types of adaptation | Incremental | Systemic | Transformational | - | - | - |
| Robert et al. (2016) [31] | Temporal scale of adaptation | Day | Season | Long-term | - | - | - |
| Regato et al. (2008) [33] | Adaptation measures | Changes in tree species composition | Conservation/restoration of biotic dispersal vectors | Changes in silvicultural practices | Changes in soil management practices | Changes in forestry guidelines | Landscape adaptation options |

| Adaptation Framework | Classification Objective | Component 1 | Component 2 | Component 3 | Component 4 | Component 5 | Component 6 |
|---|---|---|---|---|---|---|---|
| Kurukulasuriya and Rosenthal (2003) [32] | Temporal frame of adaptation measures | Short term adaptations | Long term adaptations | Adaptations irrespective of the temporal dimension of climate impacts | - | - | - |
| Smit and Skinner (2002) [18] | Adaptation measures type | Technological developments | Government programs and insurance | Farm production practices | Farm financial management | - | - |

### 1.3. Mediterranean Drylands as a Study Area

Our three case studies were developed in the south of Portugal, an area which is characterized by a Mediterranean climate, with class Csa (Mediterranean dry hot summer), which is, according to Köppen Geiger, a climate class that occupies vast areas of the Mediterranean basin and Mediterranean climate region [33,34]. According to Ramírez Villegas et al. [35], if climates are similar or analogue across space and/or time, then assessment of impacts and adaptation measures are also relevant across space and/or time. Climate scenario RCP 8.5 projects a significant decrease in rainfall and an increase in temperature and drought frequency, duration and magnitude over the Mediterranean region [36–39]. In both the south of Portugal and the Mediterranean region, some areas will change the climate classification from Csa to Bsh (Hot Semi-arid) in the climate change scenarios RCP 4.5 and RCP 8.5 [34,40]. Within these climate scenarios the fire risk will increase [41–43], and the biodiversity loss and habitat loss will increase, leaving several important plant and animal species, including trees, beyond their thresholds of survival [44–49]. This region which is, already in the present, very vulnerable to desertification [50,51] will reduce its productivity in agriculture and forestry, due to water scarcity, high temperatures, increased mortality and other impacts resultant from extreme events, such as droughts or heat waves [52–60]. Therefore, the future of agriculture and forestry in the region is, to a significant extent, threatened by climate change, making adaptation to a future climate essential in order to prevent severe ecological, social, and economic damage. The RCP 8.5 scenario for the period 2070–2100 projects a decrease in total accumulated annual precipitation in the Portuguese Alentejo region of around 20%, from 630 mm to 519 mm and in its southern area (Baixo Alentejo) from 500 mm to 400 mm [42] (Figure 1). Cork oak (*Quercus suber*) and holm oak (*Quercus ilex rotundifolia*) are the main trees in this region, forming the typical savannah-like agroforestry system called montado, and their limits for acceptable productivity are 600 mm and 500 mm of annual rainfall, respectively [61,62]. Hence, the maintenance of the cork oak and holm oak Montado landscapes [63] is at high risk. Under the RCP 8.5 scenario, for the period 2070–2100, maximum temperature is expected to rise by 4° to 5° Celsius, the number of heat waves to go from the present 40 events per year to a future 160 events, and the number of days with frost to decrease from the observed 12 days per year to a future zero or one day [42]. Due to the increase in temperature, evaporation and evapotranspiration will be higher, meaning that the demand for water and irrigation will increase, while water reservoirs will have fewer reserves in the Iberian Peninsula [56,59]. Despite this challenge, the Portuguese national climate adaptation strategy for the agriculture and forestry sector states that the territory should aim to maintain its productivity and ecosystem services in the future [64], thus clarifying the general objective of adaptation.

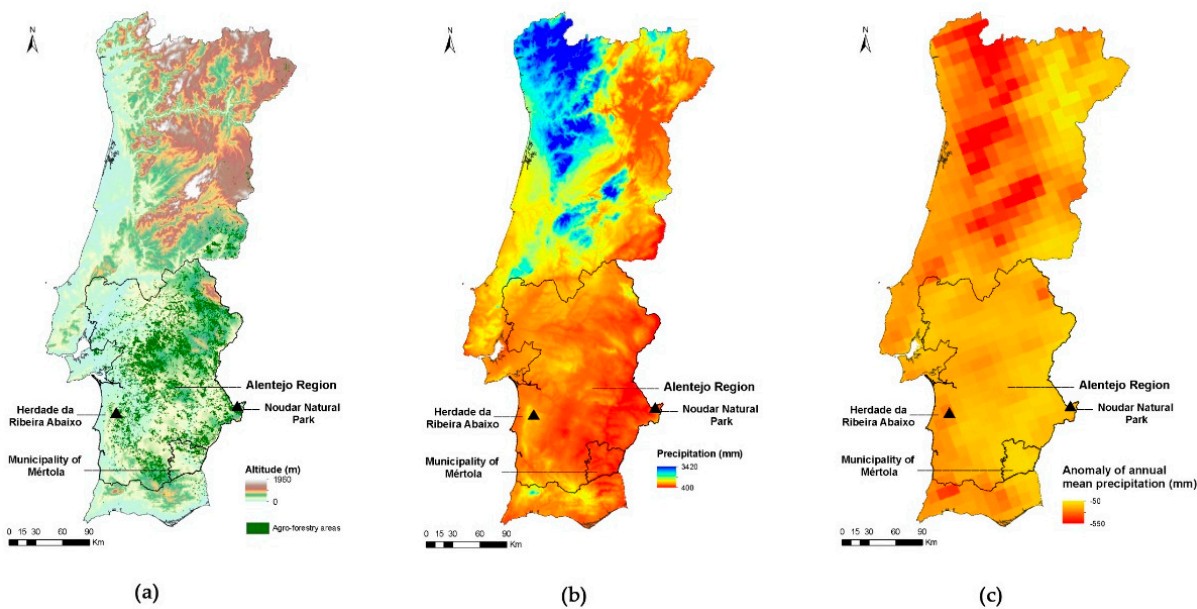

**Figure 1.** Map of Portugal with the location of the case studies of Mertola Municipality, Herdade da Ribeira Abaixo farm, and Herdade da Coitadinha/Natural Park of Noudar farm, within the study area of Alentejo region. (**a**) Altitude and agroforestry areas (from Corine land cover 2018); (**b**) observed annual mean precipitation for the period 1971–2000; (**c**) anomaly of annual mean precipitation for the period 2071–2100 under RCP8.5 scenario (datasets from IPMA, 2018).

Within this context, the present study was developed with the aim of supporting decision-making for climate adaptation of agriculture and forest landscapes in the dryland region of Mediterranean Europe, using a participatory-action-research (PAR) approach, namely at the case study region of Alentejo, in the south of Portugal. Its specific objectives are to (i) create a framework that organizes adaptation measures into strategies that are framed around their effect on climate vulnerability and to (ii) assess and compare the efficacy of adaptation measures for the reduction in vulnerability.

## 2. Materials and Methods

The methodology used to create such a framework was based on a Participatory Action Research (PAR) approach [65,66] and started with identification and clarification of the problem that arose from participatory workshops for a given study area. After identifying the problem, the authors performed an extensive review of literature on adaptation frameworks for agriculture and forestry, in order to identify possible already existing solutions. This literature review, presented in the introduction, clearly shows, and reinforces, the need for such a framework. The next step of the methodology then consisted of collecting, assembling and analysing the data and information relevant to develop a proposal. This data consists, on the one hand, of the adaptation strategies and adaptation measures themselves. This methodological step is described in Section 2.3. Review of Adaptation Measures. On the other hand, the data consists on the conceptual and theoretic frameworks available to base such a proposal on. The development of a framework based on the existing conceptual frameworks is further described in Section 2.4. Finally, this framework supported the organization of indicators, based on literature review, as explained in Section 2.5. The framework was then tested in the adaptation planning of two farms and one municipal case studies, evaluated by the participating stakeholders (see Section 2.6) and discussed by authors in the present study.

### 2.1. Study Area and Case Studies

The study was developed in the Alentejo Region, south of Portugal, with a practical application in three case studies: (i) the adaptation plan for agriculture and forestry sectors in the municipality of Mértola (130,000 ha); (ii) the adaptation plans for the Montado

agroforestry system in the Nature Park of Noudar/"Herdade da Coitadinha" farm (991 ha) and (iii) "Herdade da Ribeira Abaixo" farm (221 ha).

## 2.2. Research Stages

This study included three stages: (1) assessing the state-of-the-art of adaptation in order to create a list of measures for the adaptation to climate change of agriculture and forestry in the Mediterranean climatic region; (2) organizing measures into strategies creating a conceptual model based on cause effect relations, which can support decision-making for adaptation planning; (3) assessing the efficacy of adaptation measures.

## 2.3. Review of Adaptation Measures

To assess the state-of-the-art of adaptation measures and strategies for agriculture and forestry we implemented three methods of research. We conducted a literature review using google scholar with the keywords "adaptation", "agriculture", "forestry" and derivatives, browsing for adaptation strategies, measures, tools, and techniques for the sector. Furthermore, we looked at institutional websites [1,67,68], institutional publications, such as National Adaptation Strategies and outcomes of projects of climate adaptation using a dedicated database [68]. After this analysis we implemented a stakeholder workshop, named "Participatory State-of-the-art on Adaptation in Agriculture and Forestry of the Alentejo region", in which 43 researchers, representatives of NGOs that develop adaptation projects and the National Agency for Environment, presented their work and developed a review of the climate impacts for the region and the adaptation measures of the national adaptation strategy, from the perspective of the region [69]. Furthermore, they identified demonstration sites and farms that could be interviewed to search for more adaptation measures. We then implemented 21 semi-structured interviews to farmers that were pointed out by the three main farming federations in Portugal (Confederação dos Agricultores de Portugal (CAP), Confagri and Confederação Nacional Agricultura (CNA)) as farmers that implement good practices and are considered innovative and leaders in their farming practices. In these semi-structured interviews, we asked farmers what adaptation measures they had already implemented and what adaptation measures they would like to implement in the future.

As a result, we compiled a list of 162 adaptation measures for agriculture, forestry, agroforestry, and pastoral activities in the region (see Appendix A). Afterwards, we organized another stakeholder workshop, with 13 farmers, 8 representatives of farmers' associations/cooperatives and 15 other stakeholders (expert members of NGOs of local development or environment, representatives of a public irrigation company, national agency of environment, regional agency for irrigation support, consultants and researchers), to develop a multi-criteria analysis of the adaptation measures [70]. In this workshop, we received clear feedback that the adaptation measures and strategies needed to be organized in a hierarchical structure of strategies, measures and techniques that would support more efficient and clearer decision-making, thus reinforcing the need for this study and framework.

## 2.4. Developing the Framework for Adaptation in Agriculture and Forestry

With the objective of organizing the measures into strategies using as a base the cause effect relations that reduce vulnerability to climate change, we used the theoretical framework of the Vulnerability Framework [26], combined with the DPSIR framework [23]. The DPSIR framework Driving forces, Pressure, State, Impact, Responses was originally created to "structure and organize indicators in a meaningful way" and is presently validated as a robust tool to structure and communicate complex human environment problems and responses but also to support decision making by providing alternative effective solutions, rather than presenting predetermined solutions [71]. The use of the DPSIR framework to support climate adaptation was discussed for the first time by Eisenack and Stecker [72]. These authors consider that its main strength in this context is its structure on bio-physical

causality. Several authors discuss, on the other hand, that there is a bio-physical, but also social, vulnerability in the context of climate change. The vulnerability framework is based on the IPCC definition that climate vulnerability is a function (f) of exposure, sensitivity and adaptive capacity [26]. Adaptive capacity is the potential to reduce social vulnerability [72] and since this is a crucial aspect of climate vulnerability and adaptation, the DPSIR framework should be complemented with the vulnerability framework. The DPSIR framework was used in the past to support the climate vulnerability assessment and it was considered useful in "structuring the analysis of the linkages between cause effect relationship of vulnerability to climate change" [73].

By combining both in the analysis of the list of adaptation strategies and measures, we were able to obtain a clearer understanding of the hierarchy of adaptation strategies and the position of the different adaptation measures regarding their function in the vulnerability reduction objective (see Figure 2).

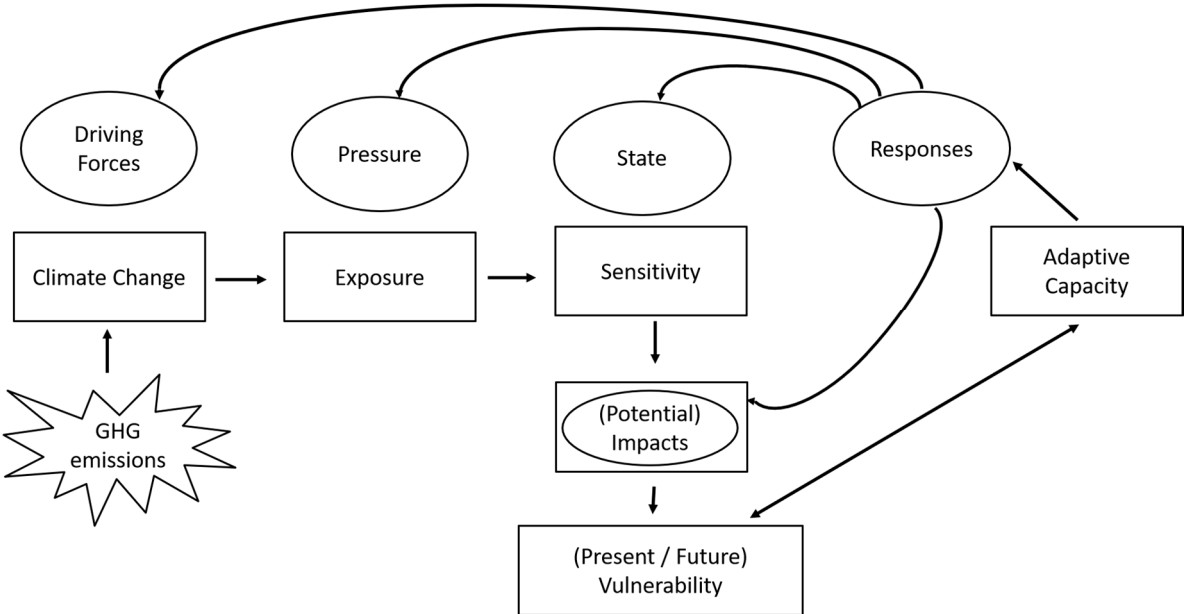

**Figure 2.** Combination of Vulnerability and DPSIR Framework. Items in rectangles refer to the Vulnerability Framework and items in circles refer to DPSIR Framework. Arrows indicate causal relations. GHG emissions are one of the main causes for Radiative Forcing and increase in Climate Change which is felt depending on the level of Exposure. Depending on the Sensitivity, this Exposure leads to Potential Impacts and Vulnerability, which is dependent on the Adaptive Capacity. Adaptive capacity can generate Responses that can be targeted at the origin of the problem, the Driving Forces and the Exposure/Pressure factors. Responses can improve the State of the ecosystem to make it less sensitive, or finally target the consequences, the Impacts on the system.

A conceptual framework proposal, naming strategies and organizing measures into strategies was then presented to colleagues and practitioners in meetings and seminars, feedback was received, and the framework was fine-tuned to be finally used to support decision-making in the adaptation planning of the three case studies. The adaptation planning of the three case studies used the resulting framework in combination with a participatory process that supported the adaptation planning, using the SWAP-Scenario Workshop and Adaptation Pathways method [11,74]. In each of these three cases studies, the framework was applied with the participation of a total of 70 participants in the six planning workshops. This participant evaluation provided feedback to support the discussion on the potential use and limitations of this study output.

### 2.5. Indicators of Efficacy of Adaptation Measures

To assess the effectiveness of adaptation measures we measured the maintenance of the productivity in the different climatic conditions. Each strategy has a different function that can be considered a specific objective, which was transformed into an indicator. The quantification of the effectiveness of the adaptation measures was then performed by a literature review, using in google scholar keywords focused on the adaptation measure and the specific indicator, for example, "mulch" + "soil moisture". The peer-review literature validated the efficacy of the adaptation measures, but only in some studies, it is quantified. This quantification is then transformed into a percentage, regarding the amount of success that the specific indicator achieves. For example, if 5 centimetres of straw mulch reduce soil annual evaporation by 38%, then this measure has the efficacy of 38%. The same measure involving another technique, using 5 cm of gravel as mulch has, on the other hand, an efficacy of 81% in maintaining soil moisture. After a thorough literature review on the list of adaptation measures and indicators, a table with the efficacy of adaptation measures was produced. This table was used in the SWAP stakeholder workshops of adaptation planning of the case studies, to support decision-making, thus receiving feedback from end-users.

### 2.6. Stakeholder Evaluation of Results

The developed adaptation framework and indicators were subject to use and evaluation in the participatory planning of adaptation of agriculture and forestry in three case studies. There was a total of 70 participant stakeholders and 55 responses in evaluation questionnaires, 14 for Herdade da Ribeira Abaixo farm, 18 for Coitadinha farm and 23 for Mértola municipality. After the workshop, the participants were asked how they evaluated the quality of the workshop of planning, its content, method and results, in a score of 1 (no opinion), 2 (bad), 3 (not sufficient), 4 (reasonable), 5 (good) and 6 (very good). The average of the evaluation of all the responses to all the case studies was 5.4. Regarding the two Vision and Planning workshops that were organised in each of the three case studies and used the framework presented in this study, 98% of stakeholders considered the presentations good or very good, 96% of stakeholders considered the content good or very good, 89% considered the final outcomes as good or very good and 95% considered the overall quality as good or very good. Furthermore, 100% of stakeholders considered workshop facilitation to be good or very good and 96% considered workshop materials to be good or very good. As a general evaluation of the whole process, participants were asked to evaluate in a score of 1 (no opinion), 2 (not satisfying), 3 (reasonably satisfying), 4 (quite satisfying) and 5 (totally satisfying). The 55 responses showed that 100% of stakeholders considered that the Method was totally or quite satisfying, and 100% of the stakeholders considered that regarding their Expectation the process was totally or quite satisfying.

## 3. Results and Discussion

### 3.1. Framework for Adaptation in Agroforestry

Based on the Vulnerability and DPSIR Frameworks, the literature review, several interviews and stakeholder workshops, we propose that vulnerability to climate change in agriculture and forests of the Mediterranean is a function (f) of several variables and can be understood and managed effectively using the following proposed framework (see Figure 3):

Climate Vulnerability = f (Potential impacts), f (Adaptive capacity).

Potential impacts = f (Exposure) [f (Climate Change Mitigation), f (Microclimates)], f (Ecosystem Resilience and Sensitivity), f (Protection).

This adaptation framework for agriculture and forestry combines the vulnerability framework, the DPSIR framework and a thorough organization of adaptation measures into strategies, based on the cause effect relation and the objectives of vulnerability reduction.

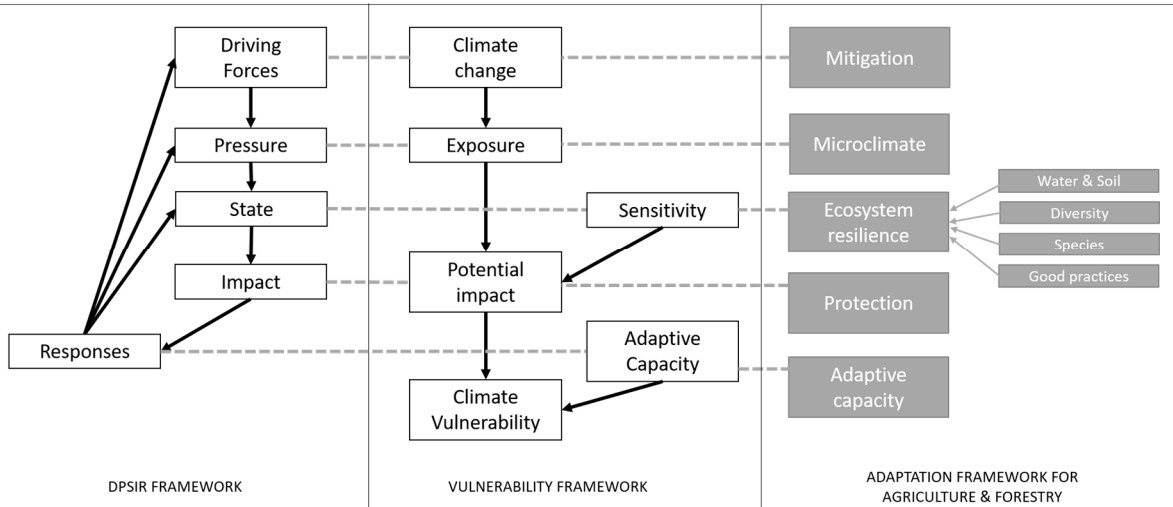

**Figure 3.** Adaptation Framework for Agriculture & Forestry. The scheme presents the interlinkages between the vulnerability framework, the DPSIR and the proposed adaptation framework. The boxes define the approach of the DPSIR framework (**left**), the Vulnerability Framework (**centre**) and the adaptation strategies in the proposed framework Adaptation Framework for Agriculture and Forestry (**right**).

Since a strategy describes how the goals will be achieved by the means, its design implies the definition of specific adaptation objectives, and depends on the understanding of the functioning of the agroecosystem and the causes for the climate vulnerability. We can divide the factors that limit or condition the productivity of plants and trees into two classes: Abiotic and Biotic. Inside the Abiotic class, we can find anthropogenic or other causes for different factors: climate, pollution, topography, water, soil and fire. Inside the Biotic class, we can also find anthropogenic and other causes or influences for some of these factors: pests, diseases, ecosystem configuration and relations, species traits, and human management of plants with techniques such as pruning, grafting, trimming, irrigation, soil management, etc.

In the context of climate action aimed to adapt the system by reducing climate induced vulnerability, based on the present framework for agriculture and forestry, we start by looking at mitigation (reducing greenhouse gases and increasing its sinks) since it addresses the cause of the problem and it is an essential strategy to reduce climate vulnerability. According to the definition in the glossary of the Fourth Assessment Report of IPCC, adaptation is the adjustment in natural or human systems response to actual or expected climate [75] and thus mitigation is exterior to the concept of adaptation. On the other hand, since the main objective of climate adaptation planning is to reduce climate vulnerability, a framework that is developed to support decision making should be clear when presenting the cause effect relations of vulnerability, thus including mitigation action in its frame. Complementarily, maladaptation actions can result in higher greenhouse emissions, thus increasing the impact of climate change in others [20] and the assessment of the risk of maladaptation by Magnan et al. conclude that the risk of maladaptation should be put "at the top of the planning agenda", namely by the frameworks that support decision making [21]. Several studies have clarified the synergies between mitigation and adaptation, namely in the agriculture and forestry sector, reinforcing the importance and opportunities in analysing efforts of mitigation and adaptation with combined approaches [76–82].

Using this framework, after analysing mitigation to reduce vulnerability, we continued by looking at how to reduce exposure to the climate pressures, then how to increase the resilience of the agroecosystem and finally, which measures are effective to protect against the direct or indirect potential impacts of the climate pressures. Complementary to these, is the strategy of investing in the Adaptation Capacity, which has to be considered as a strategy in itself due to its importance, effectiveness and direct relation to the socio-economic dimension of climate vulnerability [83].

The framework is explained bellow by presenting the adaptation strategies and a set of examples of adaptation measures for each adaptation strategy, based on the list of 162 measures (see Table A1):

1.  Mitigation—The strategy for acting on the Driving Forces of the climate-induced vulnerability, consists of Mitigation. This means focusing on actions that reduce green-house gas emissions and increase carbon sinks. If mitigation is implemented effectively at the global scale, most of the climate impacts will be minimized [84]. Even though this is outside of the typical scope of adaptation, based on the vulnerability and DPSIR framework, it must be identified as the first response that acts on the cause of the pressure. In the agriculture and forestry sector, mitigation includes measures such as preventing fires [85], increasing soil organic matter [86], afforestation [87], and the reduction in greenhouse gases production in the farm [88]. According to Smith and Olesen, measures that have positive impacts on both mitigation and adaptation include: "(1) measures that reduce soil erosion, (2) measures that reduce leaching of nitrogen and phosphorus, (3) measures for conserving soil moisture, (4) increasing the diversity of crop rotations by choices of species or varieties, (5) modification of microclimate to reduce temperature extremes and provide shelter, (6) land use change involving abandonment or extensification of existing agricultural land, or avoidance of the cultivation of new land" [82].

2.  Microclimates—According to the DPSIR framework, reducing the Pressure is the second priority when prioritizing responses. The Pressure is the Exposure to climate variables, as defined by the vulnerability framework. The Strategy of Microclimates consists of creating or using microclimates at the farm level in order to reduce the exposure to heat, cold, wind, water scarcity, etc. [89]. This strategy includes measures such as locating species on shaded North slopes (in the north hemisphere) [90], creating windbreaks [91], planting on the shade of trees or bushes [92–94], planting in riparian zones or around lakes or water reservoirs and locating species in areas with specific microclimates such as shade, sun, wind protection or wind breeze, no frost areas [95].

3.  Ecosystem Resilience—The third strategy for adaptation, combining the use of the Vulnerability and DPSIR frameworks, consists of ecosystem resilience. Resilience is defined by "the capacity of systems to absorb disturbances and still retain the same structure and function while maintaining options to develop" [96]. The resilience of the farm ecosystem depends, according to the DPSIR framework, on the State of the ecosystem, and according to the vulnerability framework, on the Sensitivity of the agroecosystem. Reducing the sensitivity and improving the state of the ecosystem will, in principle, increase the resilience of the agroecosystem. Since the framework aims to address an ecological system (agriculture) and this strategy addresses the ecological sensitivity and resilience, we have detailed it in four sub-strategies through which farm ecosystem resilience can be promoted:

    a.  Water and Soils—The net primary production (NPP) of trees or rainfed agriculture in the Mediterranean climate is mostly limited by soil water availability [97]. Increasing water availability, namely in the soil, is an important strategy that has been extensively used in the past [98] and can reduce the sensitivity of the system to droughts, reduction in precipitation and water scarcity. Measures to increase water retention in soil such as finding an optimal tree density [99], lakes, swales [100,101], terraces [102], half-moons [103], mulch [104], increase organic matter in soil [103,105] or waterboxes [106] are included in this strategy.

    b.  Diversity—Approaches based on diversity, such as agroforests, silvopastoral systems, mixed tree crop systems, multi-strata forest gardens or vegetable gardens are widespread and have been used for centuries by farmers to minimize risk and ensure some productivity in unfavourable years [79,107,108]. Diversity in the ecosystem can significantly regulate plagues, pests, fires and negative impacts on biodiversity [109]. In addition, a diversity of families, species, varieties

and genetic diversity will increase the capacity of the system to survive and prosper in different climate conditions [110–112] thus increasing its resilience [113]. Diversity can include measures such as micorrization [114] to increase survival rates; composted manure [115,116] and the plant *Phlomis purpurea* to control pests [117]; diversification of fodder using edible shrubs adapted to drought [118] or increasing species richness to increase productivity [119].

    c.    Species—The farmer or decision maker can choose the species and crops according to the present and expected future climate. A farm ecosystem with species that are adapted to the future climate variables is more resilient to climate change [19,120].

    d.    Good Practices—Promote silvicultural and pastoral practices that increase productivity while respecting the environmental carrying capacity in the long term, thus maintaining productivity in a sustainable routine. This strategy includes measures such as adequate pruning of trees, integrated rotational grazing, efficient irrigation, erosion control measures, pruning, tree protectors, fire prevention, etc. [121–125].

4.    Protection—The end of the line, but often an urgent and needed strategy, is the protection of the system elements when a certain climate change occurs. When prevention is not implemented or is not enough, this strategy can be applicable to heat waves, droughts, water scarcity, storms, or plagues in order to combat the effects of climate change impacts. This strategy includes adaptation measures such as the increase of food storage (for example hay for drought years), farm insurance, using pesticides for plague control, fire combat or irrigation [126].

5.    Adaptive Capacity—Finally, a strategy transversal to the whole adaptation process consists of increasing the adaptive capacity of the farmers and the region, therefore increasing the available set of capitals that can be used for adaptation and the capacity to mobilize them for this objective [96,127]. The capitals for adaptation include human, social, political, financial, natural, and cultural capital [127], and resources such as technology and infrastructure, information, knowledge, institutions and the capacity to learn [128]. Without adaptive capacity, all the previously mentioned strategies cannot be properly considered, assessed, evaluated, and implemented. Increasing Adaptive Capacity includes strategies such as: (a) increasing knowledge; (b) increasing financial capacity; (c) monitoring; (d) reflexive governance that can reflect and integrate the challenges of a changing system [129] and also foster the markets that can make viable the climate adapted farming [19].

The main strength that we find in this framework is, to begin with, its capacity of framing the adaptation measures and strategies regarding the main adaptation objective which is to reduce climate vulnerability. If decision-makers want to reduce the climate vulnerability of their farm or territory, they can find support and structure in this framework to understand what the strategies are and what measures they can use, either to reduce the exposure to climate change, to make their system more resilient and finally to protect against potential impacts.

This framework integrates the complexity behind the large number and type of adaptation measures, listed in the literature and proposed by several previous adaptation frameworks for agriculture and forestry, organizing them into a hierarchy of Strategies and Measures, as seen in Table 2 and in Appendix A. This hierarchization is of particular importance to the main objective of this framework is which is to support decision making by clarifying the hierarchy of adaptation measures towards the reduction in climate vulnerability. Other categories used in adaptation frameworks such as (i) types of adaptation, (ii) temporal scale of adaptation, and (iii) aim, timing, duration, scale, responsibility and form, are therefore considered complementary to this end. For further integration of these categories, other tools, such as a multicriteria table, can be used to present to stakeholders and give more information regarding the adaptation strategies and measures. Concerning the integration of adaptation domains, it is relevant to acknowledge that the strategy of

Protection includes several measures of Disaster Risk Reduction (DRR), thus supporting the integration of these two domains: DRR and Adaptation.

**Table 2.** Adaptation framework organized by strategies and a short list of adaptation measures that result from interviews, literature review, analysis and organization on cause effect relation with climate vulnerability.

| | Strategy | Strategy Short Description | Adaptation Measures (Short List) |
|---|---|---|---|
| | 1. Mitigation | Contribute to reducing greenhouse gases by carbon sequestration, reduction in energy consumption and production of renewable energy | • Prevent fires;<br>• Afforestation;<br>• Increase soil organic matter;<br>• Increase permanent pastures;<br>• Reduce machine hours;<br>• Reduce external input consumption (fertilizers, animal feed, etc.);<br>• Produce renewable energy on the farm: |
| | 2. Microclimate | Reduce exposure to climate pressure by using or creating microclimates in which there is more or less sun, wind, heat, cold or water. | • Plant trees in shade areas such as north slopes (in north hemisphere);<br>• Plant trees near riparian zones;<br>• Install windbreaks to reduce evaporation;<br>• Increase tree canopy cover to reduce heat and evaporation; |
| 3. Ecosystem Resilience | (a) Water and Soils | Optimize the relation of water demand and water availability by increasing/regulating the amount of water in the soil; increasing the water availability on the landscape by water harvesting | • Mulch;<br>• Swales;<br>• Half-moons;<br>• Terraces;<br>• Increase soil organic matter;<br>• Biochar;<br>• Lakes and dams; |
| | (b) Diversity | Using a diversity of species, varieties, crops, genes, practices, timings to create redundancy, diminish risk, create self-regulation of the ecosystem, and increase the autonomous adaptive capacity of species and landscape. | • Use diverse species and diverse genetic material to promote natural autonomous adaptation of species;<br>• Use different varieties of the same species;<br>• Install drought resistant fodder banks;<br>• Pest control through the application of composted manure;<br>• Mycorrhizal inoculation; |
| | (c) Species | Use species that are comfortable within the climate variations expected to the future, so that resistance increases. | • Choose and use species that are adapted to climate conditions namely in regard to temperature and rainfall thresholds;<br>• Use varieties that are adapted to climate conditions; |
| | (d) Good Practices | Increase the resistance of the system by improving the state of the system | • Increase the success of afforestation;<br>• Tree maintenance through pruning and trimming;<br>• Rotational grazing;<br>• Conservation tillage;<br>• Protect roots of trees from ploughing; |
| | 4. Protection | Diminish physical or socio-economic impacts by compensating the impacts with the end of the line measures | • Fire prevention and combat;<br>• Deficit irrigation;<br>• Protection of infrastructures;<br>• Insurance;<br>• Increase storage of fodder; |
| | 5. Adaptive Capacity | Increase the available set of capitals that can be used for adaptation and the capacity to mobilize them for this objective | • Rural extension;<br>• Increase and disseminate knowledge and good practices;<br>• Reduce legal and bureaucratic obstacles;<br>• Create a market for adapted crops/varieties;<br>• Articulate policies, programs and financial system to promote adaptation; |

One of the challenges that we observed when attempting to organize adaptation strategies and measures is to develop an adaptation framework that is conceptually robust and at the same time functional for stakeholders in the adaptation planning process. The organization of measures inside the presented strategies and their relation to vulnerability reduction has been well evaluated and easily accepted by stakeholders, farmers, and experts in the adaptation planning workshops in which it was used. During the participatory planning workshops, not only there positive feedback about the clarity of the adaptation strategies and measures, but also stakeholders were able to create an adaptation plan together for the territories at stake in all three case studies with the support of two workshops, thus suggesting that this organization and hierarchy between strategies and measures are satisfactory for the understanding and planning of the adaptation in the sector. The positive evaluation of stakeholders of the adaptation process supports the functional character of the framework, and further discussion can contribute to the understanding of its conceptual base, its adequate use, its limitations, and the research needed to develop future improvements.

One important attribute of this organization of measures inside strategies is that a given measure can easily fit in more than one strategy. For example, applying composted manure to soil serves, on one hand, as a Water and Soil strategy, since it increases, by 25%, the capacity of water retention of the soil [103], and, on the other hand, composted manure is a strategy of Diversity, since the diversity of microorganisms in the compost serve as a pest control, namely with an efficiency of 39–76% in controlling *P. cinnamon*, an oomycete that kills cork and holm oaks [115]. This strengthens the understanding of the different functions of adaptation measures, by clarifying the multiple positive effects of an adaptation measure by placing it in different strategies, therefore supporting different objectives. This also creates the possibility of using different indicators of efficacy for an adaptation measure for the different objective that it is trying to be achieved, as mentioned in the example above.

### 3.2. Efficacy for Adaptation in Agriculture and Forestry

In order to answer the question of "how effective is an adaptation measure", one needs to clearly identify what is the specific vulnerability reduction objective that measure is attempting to reach. The proposed adaptation framework shows that different objectives can co-exist for the success of the adaptation of a farm or territory.

Within the microclimate strategy it is possible to reduce exposure to climate variables such as solar radiation or wind and decrease temperature, decrease evaporation and increase soil moisture. It is also possible to reduce the exposure to storms and heat waves, water scarcity and frost. Within the strategy of water and soils it is possible to increase the soil moisture and therefore reduce the potential water scarcity that would result from the decrease in precipitation or increase in droughts. With the strategy of diversity, it is possible to intervene on indirect effects of climate pressure such as the productivity of the whole agroecosystem or the capacity of the species and ecosystem to respond and autonomously adapt to the potential impacts of loss of productivity that come from pest, diseases, increased evapotranspiration, heat, water scarcity and consequent limitations to growth.

Following this analysis, the specific objectives and indicators can be:

(i) Reduction in precipitation causes water scarcity. The objective is to prevent water scarcity and prevent a decrease in soil moisture. The indicator can be Soil Moisture.

(ii) An increase in temperature causes the death or reduction in productivity of certain species, when temperature rises above a given threshold. Therefore, the objective and the indicator can be Temperature regulation.

(iii) An increase in temperature and decrease in precipitation cause increased mortality and unsuccess rate in reforestation. Therefore, the objective can be the increase in the success rate of reforestation and the indicator can be the Plantation success rate.

(iv) A change in climate variables and patterns can originate more or different pests and diseases. The objective and indicator can therefore be Pest Control.

(v) An increase in the frequency, duration and intensity of Droughts can decrease the amount of fodder production for grazing animals. The objective and indicator can therefore be Fodder production during drought.

(vi) A decrease in precipitation can cause a decrease in productivity. The objective can therefore be "maintain crop productivity" and the indicator, Crop productivity.

Further objectives can be identified to reduce the vulnerability and adapt a territory or farm to climate change. If further objectives are identified, likely new indicators can and must be designed in order to confront with literature review and quantify the efficacy of the adaptation measures. The objectives and indicators mentioned above and presented in Table 3 were identified to support the adaptation planning of the three case studies and its applicability is limited by the amount of literature with quantitative studies that can provide relevant information about these adaptation measures.

**Table 3.** Efficacy of Adaptation Measures. Columns include the Strategy and the Measure of adaptation, the Indicator of Success, a description of its efficacy, a value in percentage, according to each descriptor, and, in participatory workshops, a qualitative confidence value, attributed by the planners, according to the quality of the reference and the adequacy of the reference for the context under study and planning.

| Strategy | Indicator | Adaptation Measures Used | Efficacy (%) | Efficacy Descriptor | References |
|---|---|---|---|---|---|
| Microclimate | Soil moisture content | Placement within microclimates (e.g., shaded areas; riparian zones) | 40–67% | Higher regeneration success | [90] |
| | Temperature regulation | Planting trees in the shade (e.g., under bushes) | 50% | Higher regeneration success | [94] |
| | Soil moisture content | Windbreak with vegetation | 35% | Lower evaporation within a distance from the windbreak of 4× its height | [91] |
| | Temperature regulation | Increase *Montado*'s density (to increase shade) | 40% | Shade generates 40% less heat. The decrease in 2–5 °C. The increase in natural regeneration up to 2× | [92,93] |
| Water and Soils | Soil moisture content | Mulch (ex. straw, leaf litter, stones, sawdust) | 38–81% | 38–81% water, 67% productivity | [104] |
| | Soil moisture content | Half-moon with stone walls (shape the landform to store more water) | 59–84% | Water retention, organic matter, nutrients & production (increase) | [103] |
| | Soil moisture content | Terraces (shape the landform to store more water) | 16% | 20% increase in productivity in wheat, 16% water increase | [102] |
| | Soil moisture content | Swale (shape the landform to store more water) | 2–100% | The increase in water in the soil | [100,130] |
| | Soil moisture content | Biochar in the soil | 4% | Soil water retention capacity (increase) | [105] |
| | Soil moisture content | Waterboxx or similar | 12.4%–30.2% | Increased soil moisture | [106] |
| Diversity | Plantation success rate | Mycorrhizal inoculation | 21–29% | Increase in regeneration success rate (year 1 and 2) | [114] |
| | Pest control | Pest control through the application of composted manure | 39–76% | Per cent inhibition of colony diameter of *P. cinnammomi* | [115] |
| | Pest control | Pest control through application in the soil of *Phlomis purpúrea* extract | 85% | *Phytphthora cinnammomi* control | [117] |
| | Fodder production during drought | Drought resistant fodder banks | 30%–50% | Shrub fodder biomass (increase) | [118] |
| | Fodder production during drought | Living fence with drought resistant native species | 30%–50% | Shrub fodder biomass (increase) | [118] |
| | Fodder production during drought | Biodiverse permanent pastures | 0% | Productivity increases in drought year | [131] |

**Table 3.** *Cont.*

| Strategy | Indicator | Adaptation Measures Used | Efficacy (%) | Efficacy Descriptor | References |
|---|---|---|---|---|---|
| Species | Crop productivity | Change to species adapted to climate conditions | 100% | | |
| Good Practices | Plantation success rate | Natural regeneration using individual protectors | 32–77% | Regeneration success rate (open vs shade covered) | [124] |
| Good Practices | Crop productivity | Tree maintenance through pruning and paring | 0% | Increase in acorn production | [125] |
| Protection | Crop productivity | Occasional irrigation/deficit | 89% | Almond productivity maintenance in case of drought | [126] |

The table of Efficacy of Adaptation Measures includes as columns the Strategy and the Measure of adaptation, the Indicator of Success, a description of its efficacy, and a value in percentage, according to each descriptor. The table of the efficacy of adaptation measures can be considered an application of the proposed adaptation framework. While the results in this table of efficacy include a choice of examples and literature review that fits the Mediterranean climate, other studies can be found that are more adequate to other climate regions. While the list of adaptation measures that have been identified in the literature and interviews can surpass 162 measures (see Table A1), on the table of efficacy, the number of measures is only close to 20, which reinforces the need for further literature review and research on the efficacy of adaptation measures based on these indicators and objectives.

Regarding the indicators and table of efficacy, one of its main assets is the possibility to compare the efficacy of the adaptation measures. The information on the efficacy has been considered by the stakeholders as very relevant and supportive for their decision-making. Moreover, for researchers and the project team this information has been considered very supportive to create and design the adaptation pathways for this sector, and in these specific areas.

Furthermore, when a measure is not supported by a study that assesses and quantifies its efficacy, it does not enter the table of efficacy, thus supporting the identification of knowledge gaps and the need for further literature review or need of specific research studies. The resulting table presents adaptation measures in all farm-level strategies but some of its studies of effectiveness cannot be generalized for all contexts and conditions. For example, the efficacy of mulch for water retention can be different depending on the soil structure and on the frequency of rain (and days without rain) in each region. Informing stakeholders with a general efficacy level for an adaptation measure, based on a study that only addressed specific conditions may be misleading to the extent that can misinform stakeholders into adopting an ineffective adaptation measure for their context. This raises the need for very clear communication on the limitations of each study and efficacy number. Additionally, it invites the technical team that is supporting a specific adaptation process, to review the literature, based on the same indicators, while focusing on those studies that are most relevant to the context under planning.

### 3.3. Application of the Framework in Case Studies

The number of adaptation measures in the Adaptation Framework proposed is not equal in the different strategies (see Figures 4 and 5), since it is a result of the measures identified in the literature and proposed by farmers and other stakeholders. The list should be read as a permanent work in progress as new measures can be identified from the literature or practice. Some measures can also be disaggregated into several, as they name different species or specific techniques used in different contexts. Since this framework supports analysis and decision making on adaptation, we have mapped all the adaptation measures present in the Portuguese Adaptation Strategy and the adaptation plans of the

three case studies, identifying how many measures are selected from each adaptation strategy (see Figures 4–6).

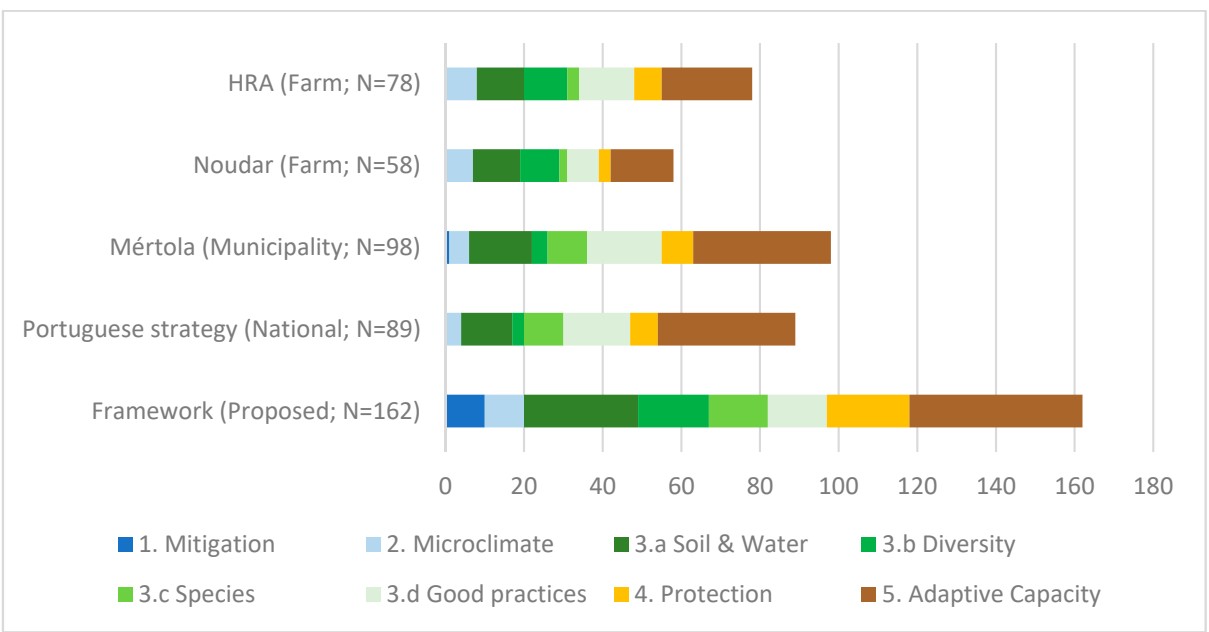

**Figure 4.** Number of adaptation measures per adaptation strategy (detailed level) identified in the adaptation plans of the three case studies, plus the Portuguese National Adaptation Strategy in comparison to the proposed Framework.

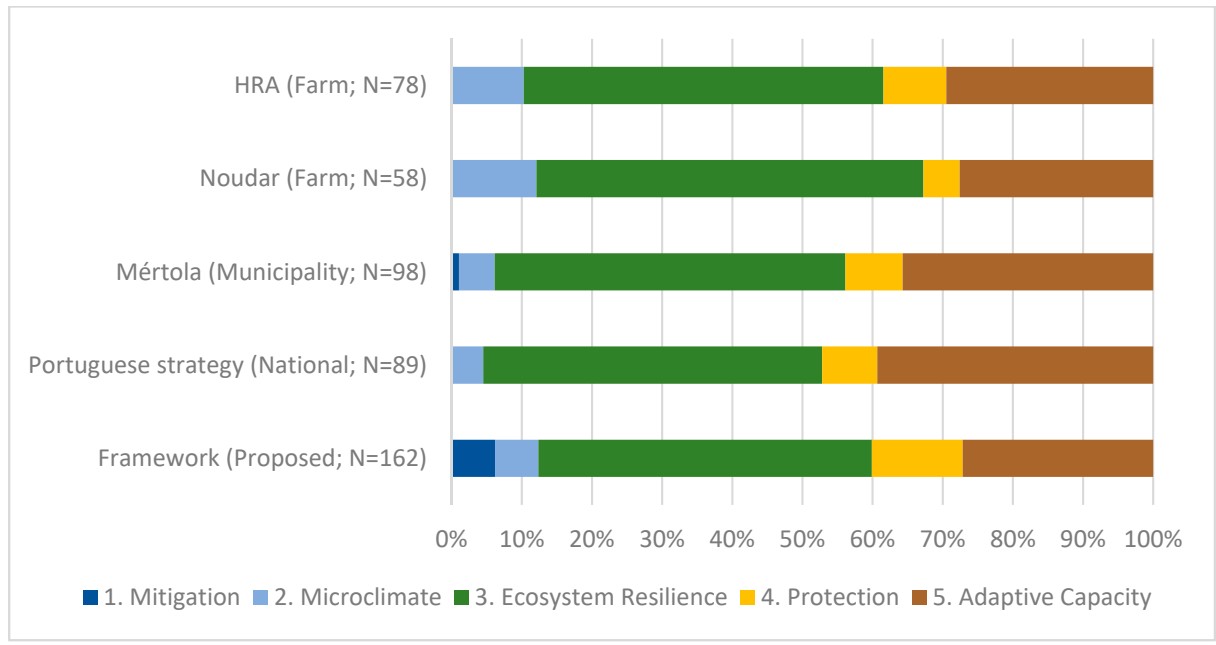

**Figure 5.** Percentage of adaptation measures per adaptation strategy identified in the three case studies adaptation plans, plus the Portuguese National Adaptation Strategy in comparison to the proposed Framework.

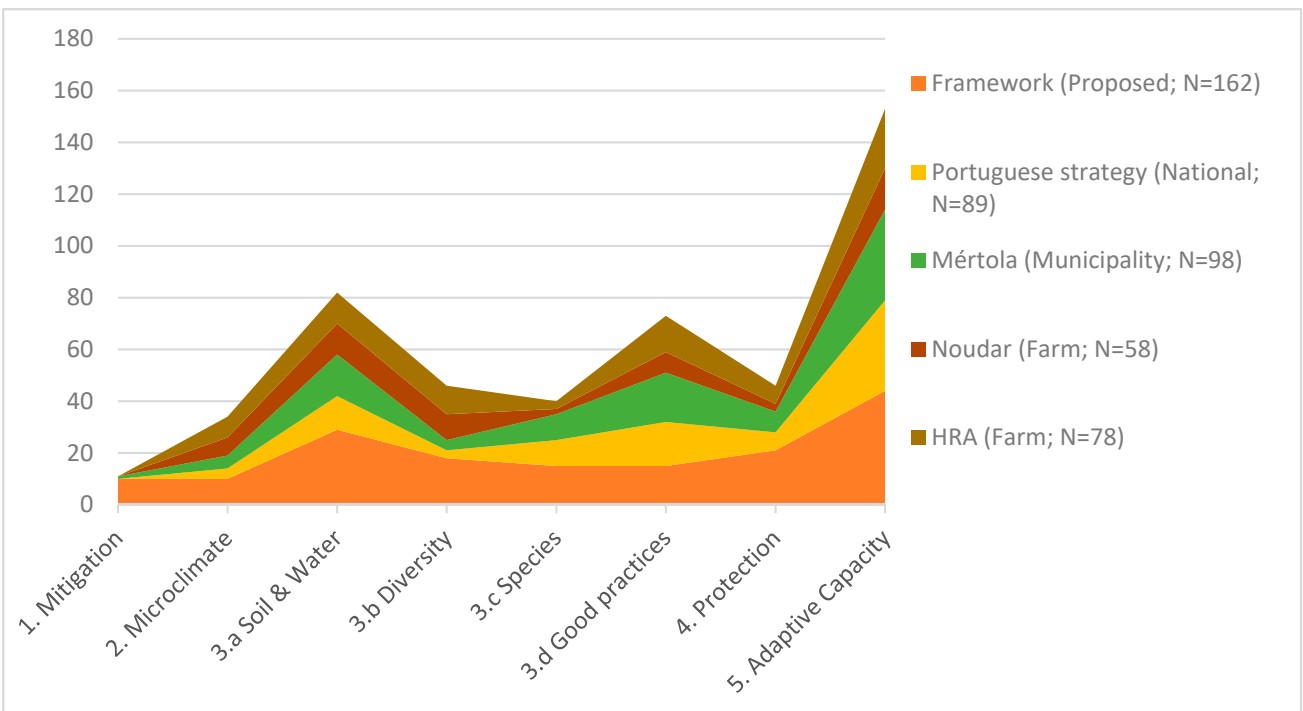

**Figure 6.** Number of adaptation measures (accumulated) identified in the case studies adaptation plans, in the Portuguese National Adaptation Strategy and in the proposed Framework, organized by detailed adaptation strategies.

The numbers of measures selected show that the Mitigation strategy was only used once and only had measures in the Mértola case study. In the Portuguese National Adaptation Strategy, this result is likely due to the fact that there are other policy instruments only dedicated to mitigation, namely the Portuguese low carbon roadmap 2050 [132]. In Mértola, the measure of installing solar panels in farms was reached by consensus in the long-term vision of the adaptation of agriculture and forestry, but, in the adaptation pathways, in the zonal plans and in the adaptive capacity measures, there is no further mention to this measure; thus, lacking more specific planning is lacking within the context of the adaptation plan for this sector. In the other case studies, mitigation measures were not selected.

Although mitigation measures are essential to reduce vulnerability, has shown by the framework, they are not selected in the adaptation plans, despite the use of this framework in the adaptation planning process. This shows the importance of having complementary instruments to plan and enforce mitigation, so that climate action and vulnerability reduction is not limited to adaptation measures. On the other hand, it is also important to notice that several of the adaptation measures selected in all case studies, namely in the strategy of microclimates, soil and water and good practices, are directly contributing to mitigation by storing carbon above and below ground, or indirectly by reducing inputs and thus GG emissions.

The analysis of Figure 5 shows that Ecosystem Resilience is the strategy with the highest percentage of measures selected, followed by Adaptive Capacity. In average 52.2% of the adaptation measures selected by the three case studies are dedicated to Ecosystem Resilience, 30.9% to Adaptive Capacity, 9.1% to Microclimates, 7.4% to Protection, and 0.3% to Mitigation. In more detail Figure 6 shows that "Soil & water" and "Good practices" are the strategies with more measures selected. This suggests the importance given to these strategies in the context of the case studies.

From the perspective of the DPSIR model, it is wise to act first on the causes and then on the consequences, which raises the question of why so few measures are used in the strategy Microclimates strategy, the one that can reduce exposure to the climate pressure. This can be due to the fact that few measures exist on this strategy, the efficacy of these measures

is only studied for few measures or their efficacy is not yet totally perceived/integrated by stakeholders. On the other hand, in future studies it would be relevant to evaluate not only the number of measures used per strategy but the extent to which they have been used in the adaptation efforts, namely in terms of financial investment.

Investing in adaptive capacity is the second most used strategy in these case studies, which is a positive result for the use of this framework, since autonomous adaptation highly depends on adaptive capacity levels and investments in adaptive capacity are considered indispensable to the success of adaptation in agriculture [133].

## 4. Conclusions

Farmers, foresters, and other agroforestry agents need to make decisions every day on their future endeavours and presently they are pressured to include climate change in their planning. Even though scientific knowledge is never complete and able to provide all the necessary recommendations for all farm activities, it is important that agricultural and forestry activities are supported with the best available knowledge to plan for climate change. The adaptation framework presented here is designed to make comprehensible, organize and effectively communicate adaptation variables, strategies and measures for the agriculture and forestry sectors. If the planning process is supported by researchers or technicians that can perform literature review on the efficacy of adaptation measures based on the presented indicators, this framework and table of efficacy can be updated and tailored to each context, thus improving common knowledge. This framework has been effective for the planning of adaptation in three case studies which shows that it has the capacity of being used in practice for adaptation planning. Stakeholders that participated in the adaptation planning evaluated the content, process, and outcomes as good and very good. With this framework it also possible to assess and compare the adaptation strategies and measures used in the different case studies, drawing conclusions about priorities and the adaptation actions that are happening in the territory. Its use in different contexts is presently under experimentation so there is space for continued improvement, for example, in the literature review on the efficacy of adaptation measures, in the further integration of the details in adaptation measures and specific objectives, and in the usability in different contexts. Its capacity to highlight knowledge gaps on the efficacy of adaptation measures to different agriculture and forestry contexts is considered relevant and supportive for the definition of future research needs. Its use for the assessment of adaptation plans and actions in different regions is also a relevant potential use and an avenue for future research.

**Author Contributions:** The first author, A.V., is the principal author of this article. All authors contributed to the study conception and design, material preparation, data collection and analysis. The first draft of the manuscript was written by A.V. and all authors contributed as joint authors in the study and manuscript. All authors have read and agreed to the published version of the manuscript.

**Funding:** This research was funded by BASE Bottom up Adaptation for a Sustainable Europe (2012–2016) (Grant Agreement No. 308337), EEA Grants/Programa AdaPT project AdaptForChange (2015–2016) and the EU LIFE Programme with project LIFE Montado-Adapt (LIFE15 CCA/PT/000043) (2016–2021). This research was also funded by Fundação para a Ciência e Tecnologia who supported the PhD grant PD/BD/113929/2015 of André Vizinho, as well as the FCT Investigator contract (IF/00940/2015) of Gil Penha-Lopes. The APC was funded by the cE3c FCT Unit funding UIDB/00329/2020.

**Institutional Review Board Statement:** Not applicable.

**Informed Consent Statement:** Informed consent was obtained from all subjects involved in the study.

**Data Availability Statement:** The data presented in this study is openly available in the reports of the adaptation plans or workshops included in this study. Should the reader require additional detail or information, it can be provided upon request to the authors. Link 1: https://www.researchgate.net/publication/332144516_Plano_de_Adaptacao_de_Mertola_as_Alteracoes_Climaticas_-_Sector_da_Agricultura_e_Florestas (accessed on 8 December 2020); Link 2: https://www.researchgate.net/publica-tion/333603290_Climate_Change_Adaptation_Plan_of_Herdade_da_Ribeira_Abaixo_farm (accessed on 8 December 2020); Link 3: https://www.researchgate.net/publication/333566708_Climate_Change_Adaptation_Plan_of_Herdade_da_Coitadinha_Noudar_Natural_Park (accessed on 8 December 2020); Link 4: https://www.researchgate.net/publication/348541537_Relatorio_do_Workshop_da_Avaliacao_Multicriterio_das_Medidas_de_Adaptacao_a_Alteracoes_Climaticas_da_Agricultura_e_Florestas_do_Alentejo (accessed on 8 December 2020); Link 5: https://www.researchgate.net/publication/336580295_Participatory_State_of_the_Art_on_Adaptation_to_Climate_Change_in_Alentejo (accessed on 8 December 2020).

**Conflicts of Interest:** The authors declare no conflict of interest. The funders had no role in the design of the study; in the collection, analyses, or interpretation of data; in the writing of the manuscript, or in the decision to publish the results.

## Appendix A

**Table A1.** Climate Adaptation Measures for the Agriculture and Forestry sector in the Mediterranean Climate. These measures are compiled from 151 different references listed below.

| #. | Strategy | | Measures | References |
|---|---|---|---|---|
| 1 | Mitigation | Carbon sink | Charcoal in soil | [105,134–136] |
| 2 | | | Increase in forest area | [137–140] |
| 3 | | | Increase in soil organic matter | [86,141,142] |
| 4 | | | Change from forage crops to permanent pastures | [143–145] |
| 5 | | Reduction of GEE emissions | Production of renewable energy on the farm | [84,146] |
| 6 | | | Reduction of enteric fermentation through diet changes | [147,148] |
| 7 | | | Decrease energy consumption on the farm (reduce tillage, fuel, fertilizer, etc.) | [149,150] |
| 8 | | | Reduction of consumption in trading (transport, packaging) | [149,151,152] |
| 9 | | | Composting of manure | [153,154] |
| 10 | | | Decrease machine hours (p.e. with conservation tilling) | [155–157] |
| 11 | MICROCLIMATES | Locate in the most appropriate microclimates | Plant in areas with specific microclimates such as shade, sun, wind protection or wind breeze, regular presence of dew and fog, no frost areas | [95,158–160] |
| 12 | | | Plant in riparian zones, around lakes or water reservoirs | [95,159,160] |
| 13 | | | Plant or use natural regeneration in shaded slopes (North faced slopes in North hemisphere) | [90] |
| 14 | | | Relocation of plants (e.g., vines) to higher latitudes (higher and cooler areas) | [161] |
| 15 | | Create microclimate to increase shade | Plant trees or bushes to create shade for other plants or trees | [92–94] |
| 16 | | | Create shade (e.g., by not clearing shrubs completely (make strips or stains) or plant bushes or trees for shade) | [92–94] |
| 17 | | | Increase the density of the montado (to increase shade) | [93,160] |
| 18 | | Create microclimate to lower temperature | Lower air temperature with creation of water bodies (e.g., lakes); creation of a phytoclimate with trees or plants; lower soil temperature with mulch or shade. | [158,162,163] |
| 19 | | Create microclimates to increase water in the soil | Creation of windbreak with vegetation | [164] |
| 20 | | | Afforestation (with trees with low water use) to increase air moisture, rain, dew and reduce soil moisture loss. | [165–167] |

**Table A1.** *Cont.*

| #. | Strategy | Measures | References |
|---|---|---|---|
| 21 | | Half-moons/boilers in boomerang around the trees | [103] |
| 22 | | Half moons with stone walls | [103] |
| 23 | Preparation and modelling of terrain to increase water retention | Terraces (Model the terrain to store more water) | [102] |
| 24 | | Swales | [100,101] |
| 25 | | Plantation in contour | [168,169] |
| 26 | | Plantation and Mobilization in keyline | [170] |
| 27 | | Waterboxx or similar | [106] |
| 28 | | Tilling on the line | [171] |
| 29 | Conservation tilling | Direct seeding | [172,173] |
| 30 | | Conservation tilling | [174,175] |
| 31 | | Non-tilling | [176] |
| 32 | | vegetable cover with green manure | [177,178] |
| 33 | | Mulch | [104,179] |
| 34 | Increase the water retention and improve the soil with vegetation | Chop and drop: chop the herbs after the last rains to get more organic matter protecting the soil | [177,180] |
| 35 | | Barriers of bush or vegetation in contour | [168,181] |
| 36 | | Manure | [178,179] |
| 37 | | Create temporary ponds | [98,182] |
| 38 | Increase the water storage capacity | Create permanent Ponds/Lakes | [98,182] |
| 39 | | Irrigation from large dams | [98,182] |
| 40 | | Feeding of groundwater and aquifers | [183,184] |
| 41 | | Drip Irrigation | [185] |
| 42 | | Use water-efficient irrigation systems and practices | [185] |
| 43 | Improve the watering efficiency | Use weather forecast for agricultural activities | [186] |
| 44 | | Monitoring the amount of water required for watering with probes | [187] |
| 45 | | Fertilization of soil with living organic matter | [179] |
| 46 | Improve the soil with additives | Charcoal / biochar in soil | [105] |
| 47 | | Control soil pH and nutrients with additives | [179] |
| 48 | | Placement of treated sludge in the soil | [188] |
| 49 | Diversify the sources of water | Reuse of wastewater for irrigation | [189] |
| 50 | Use forage shrubs resistant to drought | Living fence with native species resistant to drought | [190,191] |
| 51 | | Drought-resistant forage banks | [192,193] |
| 52 | | Preservation of wild fauna and flora | [194,195] |
| 53 | | Diversify cultures/crops | [196] |
| 54 | Greater diversity of the type of culture, species, varieties and genes | Diversifying species and uses of soil | [119,197,198] |
| 55 | | Use of biodiverse permanent pastures | [131,199] |
| 56 | | Diversification of varieties (drought resistance or to take advantage of the anticipation of phenology) | [197,200] |
| 57 | | Organic farming practices for biological control of pests | [201] |
| 58 | Use biodiversity to control plagues | Increase the presence of insectivorous birds for pest control | [201] |
| 59 | | Creation of biodiversity hotspots for insect balance | [109] |
| 60 | Increase the diversity and complexity of the agro-ecosystems | Inoculation with mycorrhizal fungi | [114,202] |
| 61 | | Conservation/regeneration of riparian zones | [109] |
| 62 | Increase the diversity and complexity of the agro-ecosystems | Increase the biological complexity of the forest system | [109,203] |

SOIL & WATER

DIVERSITY

Table A1. *Cont.*

| #. | Strategy | Measures | References |
|---|---|---|---|
| 63 | Create or maintain silvo-pastoral-systems | Create new complementary products from the Montado | [113,204] |
| 64 | | Value the complementary products of the Montado | [205] |
| 65 | | Maintain agro-silvo-pastoral system | [206] |
| 66 | Increase the diversity with exotic species | Complementary products of Montado (exotic species) | [113,207] |
| 67 | | Agro-silvo-pastoral system with exotic species | [208] |
| 68 | Selection of species according to predicted climate conditions | Switch to better adapted species | [209,210] |
| 69 | | Choose better adapted varieties, with more adequate thermal needs and more resistant to thermal and water stress | [211–213] |
| 70 | | Abandon cereal farming and adopt other crops such as forestry | [214] |
| 71 | | Use of more rustic and adapted animal species | [215] |
| 72 | | Use of short cycle species (annuals) to reduce watering | [216] |
| 73 | | Extension of the production period by the use of earlier or later cultivars | [216] |
| 74 | Increased genetic diversity in crops | Greater and better genetic diversity in crops/species | [217,218] |
| 75 | | Use of locally adapted varieties (either by local seed or selected clones) | [217,218] |
| 76 | | Use local and indigenous cereal varieties | [219] |
| 77 | | Creation of local/regional seed banks | [219] |
| 78 | | Preservation of intervarietal and intravarietal biodiversity in olive trees | [64,220] |
| 79 | Selection and improvement of species | Species selection and improvement | [64,217] |
| 80 | | Genetic improvement program: selection of cultivars adapted to thermal stress, drought, etc. | [64] |
| 81 | | Collect and use seeds from the best plants from the most adverse locations | [221,222] |
| 82 | | Installation of rootstocks more resistant to water shortage | [223] |
| 83 | Soil | Increase soil organic matter | [141] |
| 84 | | Promote and maintain mycorrhizae in soil | [202] |
| 85 | | Crop rotation | [224] |
| 86 | | Fertilization with compost (e.g Bokashi method) | [225] |
| 87 | Forestry | Individual Protectors of natural regeneration | [124] |
| 88 | | Tree maintenance with pruning | [125] |
| 89 | | Cork Oak climax forest in sloping areas | [33] |
| 90 | Animals | Soil tilling with pigs | [226] |
| 91 | | Integrated management of grazing to promote forest regeneration and clearing of weeds | [196] |
| 92 | | Integrated and intensive rotation of livestock (Holistic management) | [226,227] |
| 93 | | Feed livestock with sprouted cereal | [228] |
| 94 | Fruticulture/Vinyard | Use sturdy rootstocks | [223] |
| 95 | Annual crops | Adjust the date of sowing/planting according to the thermal regime of each year to extend the production cycle | [18] |
| 96 | | Use of the best practices of rainfed agriculture | [122,229] |
| 97 | | Make two irrigated crops in the same year (due to increased heat) for cattle feeding | [18,230] |

SPECIES (rows 68–82), GOOD PRACTICES (rows 83–97)

**Table A1.** *Cont.*

| #. | | Strategy | Measures | References |
|---|---|---|---|---|
| 98 | PROTECTION | Protect crops from water scarcity | Viticulture: Changes in cultural practices and driving systems, namely to optimize/reduce water consumption by the crop, increasing the efficiency of water use | [200] |
| 99 | PROTECTION | Protect crops from water scarcity | Anticipation or delay of sowing according to the climate | [18] |
| 100 | PROTECTION | Protect crops from water scarcity | Punctual/ deficit irrigation | [189,231] |
| 101 | PROTECTION | Protect crops from water scarcity | Permanent irrigation | [232,233] |
| 102 | PROTECTION | Protect the farm against floods | Promote the cleaning and normalization of water lines, involving technical training and taking into account the maintenance of riparian vegetation | [234,235] |
| 103 | PROTECTION | Protect the farm against floods | Natural Flood Management | [236,237] |
| 104 | PROTECTION | Increase forage stocks (in good years) | Increase hay and straw stock (in good years) | [18] |
| 105 | PROTECTION | Fire protection | Improve Forest Fire Management (Fire Prevention; Fire Detection; Initial attack; Fuel management) | [236] |
| 106 | PROTECTION | Fire protection | Fire breaks; Fuel breaks and Green belts | [237] |
| 107 | PROTECTION | Fire protection | Promoting heterogeneous agro-forest mosaics | [238] |
| 108 | PROTECTION | Fire protection | Strategic Forest Planning at Landscape level | [239–241] |
| 109 | PROTECTION | Protect from storms and strong winds | Increase the strength of greenhouses and structures | [242] |
| 110 | PROTECTION | Protect from storms and strong winds | Use stronger tutors for plants | [18] |
| 111 | PROTECTION | Protect crops and animals from heat waves | Use sprinkler and fogging to reduce the temperature | [242] |
| 112 | PROTECTION | Protect crops and animals from heat waves | Install artificial shade | [242] |
| 113 | PROTECTION | Protect crops and animals from heat waves | Strengthening of environmental control equipment in protected cultures (cooling's, etc.) | [242] |
| 114 | PROTECTION | Fight plagues and diseases | Fight plagues with application of plant based pesticides | [117,243] |
| 115 | PROTECTION | Fight plagues and diseases | Fight plagues (e.g., P. Cinnamomi) with manure application | [115,116] |
| 116 | PROTECTION | Fight plagues and diseases | Installing pest traps | [244] |
| 117 | PROTECTION | Protect crops from water quality degradation | water treatment or irrigation water (e.g., constructed wetlands) | [245] |
| 118 | PROTECTION | Agricultural insurances | Agricultural insurance against extreme events and production losses | [246] |
| 119 | ADAPTIVE CAPACITY | Promotion and Training | Rural extension/agricultural advice linked with training/research/demonstration | [19,69,123,129, 247,248] |
| 120 | ADAPTIVE CAPACITY | Promotion and Training | Training of public administration and private sector technicians in this area | [19,69,123,129, 247,248] |
| 121 | ADAPTIVE CAPACITY | Promotion and Training | Rewarding Early Adopters | [19,69,123,129, 247,248] |
| 122 | ADAPTIVE CAPACITY | Promotion and Training | Documenting and disseminating good traditional practices | [19,69,123,129, 247,248] |
| 123 | ADAPTIVE CAPACITY | Promotion and Training | Environmental education (e.g., in schools) | [19,69,123,129, 247,248] |
| 124 | ADAPTIVE CAPACITY | Promotion and Training | Produce and disseminate more practical and useful knowledge | [19,69,123,129, 247,248] |
| 125 | ADAPTIVE CAPACITY | Promotion and Training | Establishment of demonstration centres for good practices | [19,69,123,129, 247,248] |
| 126 | ADAPTIVE CAPACITY | Promotion and Training | Promotion of landscape connectivity for species migration | [19,69,123,129, 247,248] |
| 127 | ADAPTIVE CAPACITY | Research and development | Increase knowledge of climate change scenarios | [19,64,69,96] |
| 128 | ADAPTIVE CAPACITY | Research and development | Evaluation of new cultivars more adapted to climate change | [19,64,69,96] |
| 129 | ADAPTIVE CAPACITY | Research and development | Classification of species and varieties in groups according to their vegetative cycle and resistance to climatic factors | [19,64,69,96] |
| 130 | ADAPTIVE CAPACITY | Research and development | Increase knowledge of the effectiveness of adaptation action (actions and different contexts) | [19,64,69,96] |
| 131 | ADAPTIVE CAPACITY | Research and development | Develop simpler technologies for the exploration of species of natural resources more adapted to the future | [19,64,69,96] |
| 132 | ADAPTIVE CAPACITY | Research and development | Promote applied and interdisciplinary research | [19,64,69,96] |
| 133 | ADAPTIVE CAPACITY | Research and development | Creation of diagnostic tool to assess adaptation needs | [19,64,69,96] |

**Table A1.** *Cont.*

| #. | Strategy | Measures | References |
|---|---|---|---|
| 134 | | Study of measures to adapt to situations of heat (extreme heat) involving producer organizations and the scientific community | |
| 135 | | Improvement of the meteorological system for warnings farmers in case of such events | |
| 136 | | Strengthening of the insurance system, particularly at the level of agricultural installations and production | |
| 137 | | Development of studies that identify areas of greater vulnerability to extreme events with proposals to mitigate this phenomenon | |
| 138 | | Development/improvement of knowledge on indicators of the water status of crops | |
| 139 | | To deepen the knowledge of the genetic diversity of the forest, plant and animal species, to promote the long-term conservation of a broad genetic base and to ensure the availability of genetic heritage and the production of reproductive material with the characteristics and diversity appropriate to the needs of the sectors, considering the expected impacts of climate change | |
| 140 | ADAPTIVE CAPACITY / Policy | Changing the Decision-Making Environment (under which management-level adaptation activities typically occur) | [19,64,69] |
| 141 | | Compatibility/articulation between national and community policies and different territorial management instruments | |
| 142 | | Negotiating the inclusion of drought situations in the context of the implementation of the EU Solidarity Fund | |
| 143 | | Reinforce alert systems and create prevention and emergency procedures | |
| 144 | | Reinforce the mechanisms and instruments needed to improve forest management and reduce abandonment | |
| 145 | | Promote the carbon sequestration capacity of forest and agricultural ecosystems | |
| 146 | | Promote diversification of products in forestry and farms | |
| 147 | | Support systems for certification, promotion and marketing of products with differentiated quality as well as innovation in this area | |
| 148 | ADAPTIVE CAPACITY / Governance | Promote local systemic vision in agricultural and regional planning | [19,64,69] |
| 149 | | Reinforcing the mechanisms and instruments needed for forest improvement | |
| 150 | | Keeping population in rural areas | |
| 151 | | Promoting access to land and renewing farmers | |
| 152 | | Strengthening the role of agriculture and forestry in protecting soil and water | |
| 153 | | Adapt the governance system to the vision | |
| 154 | ADAPTIVE CAPACITY / Financing | Pay farmers, pastoralists and foresters for their services to the ecosystem and pay according to the services provided | [19,64,69] |
| 155 | | Develop and financially support investment in irrigation | |
| 156 | | Support investment in, specifically, more efficient irrigation systems, improved management and irrigation warning systems | |
| 157 | | Creation of support for the maintenance of native species and breeds as well as traditional varieties | |
| 158 | | Financial support for prevention actions and reimbursement of damages | |
| 159 | | Creation of support for the maintenance of native species and breeds as well as traditional varieties | |
| 160 | Monitoring | Create environmental impact alert systems (using impact and non-effect indicators) | [19,64,69] |

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
