# Peer review of "Framework for Climate Change Adaptation of Agriculture and Forestry in Mediterranean Climate Regions"

_land, doi:10.3390/land10020161_

Round 1
Reviewer 1 Report
This manuscript try to consider the adaptation of agriculture and forestry sector with different scale and including several stakeholders based on the framework from various literatures. It is very interesting points for the authors to consider the adaptation measures based on developed framework in Portugal.
One of challenging points of this manuscript, however, is unclear structure, particularly IMRAD. It is very difficult to understand what the problem statement of this manuscript is. In addition, the authors did not clearly state the objectives as well. And unfortunately, I cannot find out methodology section in this manuscript. Please clearly state the problem statement, objectives and methodology in this manuscript. In related with problem statement, the authors should clarify the challenges of adaptation in agriculture and forestry sector and explain why a framework is necessary(or useful).
In addition, this manuscript have several points to be developed. Please develop the manuscript based on the comments described as bellows.
Abstract
Please state clearly problem statements and objectives of this manuscript.
Line 64
The authors state these three “the crop, the cropping system and the farming system”. How different these scale are ? It is not so clear how diffident. Please clarify this.
Line67-68
Please change “Howden and colleagues(2007)” into “Howden et al. (2007)”
Line 196
Please unify same unit either “ha” or “km2” in three different research sites.
Line 238-(section 2.4)
Why and how do the authors select the Vulnerability Framework, combined with the DPSIR framework into your research.
Line 333
According to this ,it seems that mitigation include adaptation. But normally, it does not. What do the authors thing this point.
References
The some reference information should be revised. Please check the references contents in attached manuscript.

Author Response
Dear Reviewer,
Thank you very much your attention in revising our manuscript and the very concrete and detailed comments. Please see the attachment for our point to point explanation of our corrections and updates. We also update the new version of the manuscript.

Reviewer 2 Report
The paper presents a highly relevant framework and tool for climate change adaptation.The authors build on a thorough basis of state of the art regarding existing frameworks and offers a developed 'pilot' version of a new framework. It is a clear strength that the framework is also tested on case studies and with convincing stakeholder participation. In my view the paper is ready for publication, however a few comments could be considered to add clarity:
The tables are complex. This is perhaps a necessary consequence of the holistic approcah in developing the framework. However, Table 1 is difficult to grasp. More specifically, even if the text after the table explains what the table is about it is hard from the table itself to see the same. So please have another look on the table.
Under 2.3 and line 228 it is said that the questionnaire asked farmers about adaptation measures among other things. I would like to know what 'other things' were addressed in the interviews. Line 232-233 there is mention of a workshop. Is this a different workshop from the one mentioned in line 219, and if that's the case - who participated?
Under 3.1, line 308-309 what does the 'f' stand for? In Figure 3, there are two boxes with 'Adaptive capacity', this is somewhat confusing and should be better explained in the following text.
Author Response
Dear Reviewer,
Thank you very much for your attention in revising our manuscript and the very concrete and detailed comments. Please see the attachment for our point to point explanation of our corrections and updates. We also update the new version of the manuscript.

Round 2
Reviewer 1 Report
Dear the authors,
When I read this revised manuscript, I found that the quality of this manuscript has been improved drastically. I think this manuscript almost reach to publish for "Land".
I have only minor comment in the references.
References
The authors almost revise the style correctly, thought I found only some points. See yellow marker as attached your manuscript. See attached your attached file.
